# Siblings and nonparental adults provide alternative pathways to cultural inheritance in juvenile great tits

Sonja Wild 🆔[1,2,3]*, Gustavo Alarcón-Nieto[1,4,5,6], Lucy M. Aplin[1,7,8]

1 Cognitive and Cultural Ecology Research Group, Max Planck Institute of Animal Behavior, Radolfzell, Germany, 2 Centre for the Advanced Study of Collective Behaviour, University of Konstanz, Konstanz, Germany, 3 Department of Environmental Science & Policy, University of California Davis, One Shields Ave, Davis, California, United States of America, 4 Department of Migration, Max Planck Institute of Animal Behavior, Radolfzell, Germany, 5 International Max Planck Research School for Quantitative Behaviour, Ecology and Evolution, Konstanz, Germany, 6 Department of Biology, University of Konstanz, Konstanz, Germany, 7 Department of Evolutionary Biology and Environmental Studies, University of Zurich, Zurich, Switzerland, 8 Division of Ecology and Evolution, Research School of Biology, Australian National University, Canberra, Australia

* swild@ucdavis.edu

## Abstract

In many animal species, the juvenile period is under strong selection, leading to a concentration of social learning during this stage as an efficient strategy for young individuals to acquire skills essential for survival. However, as social learning is not always adaptive, juveniles need to be strategic in when, who, and what to copy. In species with extended parental care, parents are often preferred sources of information, leading to stable intergenerational transmission of knowledge. However, little is known about transmission pathways in species with limited periods of parental care, and their implication for cultural inheritance. Here, we investigate social learning strategies during development in a model species with a dependence period of a few weeks, the great tit (*Parus major*). Using fully automated two-option foraging puzzles, we diffused knowledge about the puzzle through breeding populations and then constrained parental individuals' choices such that parents either (1) both had knowledge of the same option, (2) had conflicting knowledge of the two options, or (3) had no knowledge of how to solve the puzzle. We then tracked solving behavior of 229 newly fledged juveniles over 10 weeks. Parental solving frequency during dependence strongly predicted knowledge acquisition by offspring, suggesting intergenerational cultural inheritance. However, detailed investigation of learning pathways revealed siblings as the most important role models for social learning, followed by nonparental adults and parents. Furthermore, offsprings' option choices were not predicted by parental choices, but instead influenced by the broader social environment, with evidence for a conformist learning bias. Overall, by using large-scale experimental manipulation of parental behavior, our study offers new insights into social learning

**Data availability statement:** Data and R code to replicate analyses and re-create figures can be found at https://doi.org/10.5281/zeno-do.16930533 .

**Funding:** This work was supported by the Deutsche Forschungsgemeinschaft (DFG, German Research Foundation - https://www.dfg.de/en/research-funding/funding-initia-tive/excellence-strategy) under Germany's Excellence Strategy (grant number EXC 2117—422037984) and a Max Planck Society Group Leader Fellowship (https://www.mpg.de/max-planck-research-groups) to LMA. LMA is cur-rently supported by the Swiss State Secretariat for Education, Research and Innovation (SERI—https://www.sbfi.admin.ch/en) under contract number MB22.00056. SW was sup-ported by a Swiss National Science Foundation postdoc mobility fellowship (https://www.snf.ch/en/XIZpfY3iVS5KRRoD/funding/careers/postdoc-mobility) during preparation of this manuscript (grant number P500PB_210994). The funders had no role in study design, data collection and analysis, decision to publish, or preparation of the manuscript.

**Competing interests:** The authors have declared that no competing interests exist.

**Abbreviations:** ILVs, individual-level variables; NBDA, network-based diffusion analysis; OR, odds ratio; PIT, passive integrated transponder.

pathways and mechanisms of cultural inheritance in r-selected species with limited parental care and multiple offspring. Our findings provide a stark contrast to most previously studied systems exhibiting multigenerational cultures, where cultural trans-mission overwhelmingly occurs from parents to offspring, and give insights into the more variable transmission routes that might occur across socially learning species.

## Introduction

In many animal species, the juvenile period is critical for acquiring skills necessary for survival, including knowledge about diet [1,2], foraging techniques [3], predator avoid-ance [4,5], migration routes [6,7], and how to socially interact with conspecifics [8]. Social learning, i.e., learning that is influenced by observation of or interaction with conspecifics [9], can greatly increase the efficiency with which individuals acquire novel information or behavior. For efficient acquisition of behavior, reliance on social learning is therefore often particularly pronounced during the juvenile period [10]. The adaptive benefit of social learning during the juvenile period has been argued to lead to selection on family living and parental care, with the cultural inheritance of knowl-edge from parent to offspring in turn leading to selection for extended juvenile periods and cognitive evolution [11,12].

Thus, many species with prolonged periods of parental care rely extensively on so-called vertical learning between parent and offspring, e.g., [13], leading to patterns of behavioral similarities within family units (e.g., Bornean orang-utans [1], vervet monkeys [14], and dolphins [3,15]). This then in turn leads to the intergenerational cultural transmission of knowledge, promoting stable multigenerational cultures with adaptive significance and ecological implications [16]. Yet, social learning is not restricted to species with prolonged dependence periods. This is particularly true in many bird species who are typically characterized by short durations of parental care [17]. The learning strategies of developing individuals in species with limited periods of dependency are less well understood, and information on how such strategies might change during transition to independence is lacking (but see [18,19]).

The vertical transmission of behavior from parent to offspring may not always be adaptive, especially in rapidly changing environments or in environments that differ over fine spatial scales, where dispersing juveniles may experience different condi-tions to their parents. Here, young individuals may need to be strategic in *when* to rely on social learning. The value of parental social information is expected to rapidly decrease as young individuals gain independence and experience new environ-ments, in which they may benefit from other role models who may be more familiar with the local environment [20,21]. Furthermore, individuals also need to be strategic in *who* to choose as role models, as conspecifics may differ in the value of infor-mation they hold [22–24], and parents may already possess outdated knowledge. Finally, as transmission of information or behavior can only occur among individuals who are in close enough proximity for learning to occur, a species' social system and social network structure will strongly influence the social learning strategies that

developing individuals employ, with limited parental care potentially favoring learning from siblings or other peers. However, the effect of such social learning strategies on the potential for multigenerational cultures is not well understood.

Here, we conduct a comprehensive investigation into the ontogeny of social learning in a species with a comparatively short period of dependence, the European great tit (*Parus major*). Great tits have become a model species for social learning and animal culture, with extensive evidence that both foraging behaviors and song can be socially learned [25,26], with new foraging behaviors spreading rapidly through social networks to establish as multigeneration cultural traditions [27]. This body of evidence suggests that individuals can learn foraging behaviors throughout life via oblique or horizontal (peer to peer) learning from flock-mates and exhibit adaptive social learning strategies when doing so. For example, they may exhibit a conformist learning bias when learning new behaviors [27], but shift to a pay-off learning bias when moving into new environments with different foraging behaviors [28]. However, almost all studies on foraging behavior have been conducted in the winter flocking period when first-year juveniles are already fully independent. We currently lack information on how social learning is expressed during ontogeny, a time marked by dynamic changes in association patterns with parents, siblings, peers and other adults [29], and with it, shifting availability of potential role models for social learning.

Great tits are seasonal cavity breeders, with parents raising their offspring in the nest for approximately 22 days [30]. After fledging, offspring spend between 10 and 32 days in their family group, during which they are fed by parents [31]. With the onset of independent feeding, they reach full independence from parents and integrate into local flocks, but may continue to prefer to associate with siblings and peers over adults [29]. Previous cross-fostering experiments, where eggs were swapped between great tits and blue tits (*Cyanistes caeruleus*), found that for behaviors related to mate choice, foraging behavior and song, birds showed more similarity to their foster parent species than their biological parent species, indicating the these behaviors are learned rather than genetically determined [32–35]. However, the mechanisms of transmission remain unresolved. On the one hand, it is plausible that offspring could have learned these behaviors directly from their foster parents through vertical learning. Alternatively, however, developing individuals could have imprinted on the "wrong" species during a sensitive period and subsequently copied behavior from other individuals in their foster species other than their foster parents after reaching independence.

To examine the ontogeny of social learning in wild great tits, we conducted a social learning experiment using two-option foraging puzzles with a sliding door that have been previously used in social learning experiments in this species, leading to multigenerational cultures [27,36]. To obtain a mealworm reward, birds could slide a door either to the left or the right [37]. We introduced and diffused knowledge of these puzzles in two populations of breeding adults and then used a selective access mechanism [37] to constrain individuals in their side options. This resulted in four experimental conditions for breeding pairs: (1) parents with the same knowledge state (both left or both right), (2) parents with conflicting knowledge states (left and right), (3) parents with only one knowledgeable adult (left or right), and (4) parents without any knowledge. We then examined the acquisition of knowledge by their offspring in the immediate period after they fledged from the nest. First, to explore to what extent offspring behavior is predicted by vertical transmission from the parents, we asked whether parental solving frequency during dependence—irrespective of side choices—increased the likelihood of offspring learning to solve the puzzle. Second, we investigated the detailed pathways through which juveniles learn to solve in a "network-based diffusion analysis" (NBDA) [38–40], to establish the relative importance of vertical, oblique, and horizontal learning. Finally, we investigated the decision-making strategies juveniles employed when choosing between the two options (solving right or left), the influence of parental solving behavior on these strategies, and how these strategies change with increasing personal experience.

## Results

Both adult great tits and nestlings in two breeding populations (MI and GU) were equipped with unique passive integrated transponder (PIT) tags for remote identification using RFID. Across both populations, we monitored 306 nest boxes, from which 51 breeding pairs successfully fledged 229 chicks.

 

PLOS Biology

For the social learning experiment, we deployed seven automated puzzle boxes with a sliding door and diffused solutions through population for two weeks (Fig 1A). Solvers were identified via RFID antennae built into the landing perches. Of the 51 breeding pairs, 26 were both knowledgeable in solving: 13 were restricted to solve on the same side (eight right, seven left) and 13 to opposite sides. Two pairs had one knowledgeable partner each (both right), and 23 pairs had no knowledge. The numbers of parents with different knowledge states reflect both our experimental design, which prioritized exposing juveniles to solving parents, and natural variation in survival and site use, which limited which breeding pairs were ultimately included in the analyses.

After fledglings emerged from their nest boxes, puzzles were deployed for an additional eight weeks on four days per week, recording identity and side choices of both solving adults and juveniles. Juveniles were considered knowledgeable if they produced at least 10 solves.

On two days per week, we additionally deployed seven RFID-equipped feeder columns ("network feeders") to track associations based on cofeeding events. In total, 231 tagged great tits—of which 103 were juveniles—were recorded (MI: 170 [Fig 1B]; GU: 61). Of the juveniles, 53 were knowledgeable of the puzzle box, producing 10,668 solves across the experiment.

## Influence of parental behavior during dependence on solving

In a first analysis, we investigated to what extent solving behavior is predicted by vertical transmission from parents, specifically testing whether more frequent exposure to solving behavior during dependence increased the probability of whether juveniles learn to solve. In a Bayesian logistic regression [41], we tested whether the log number of parental solves—regardless of side choices—or log number of parental scrounges during dependence predicted whether offspring learned to solve the puzzle or not. Scrounging occurs when a bird flies in immediately after a solve occurs and picks out a worm before the door fully closes, representing an alternative strategy to solving to obtain a food reward [42]. Inclusion of

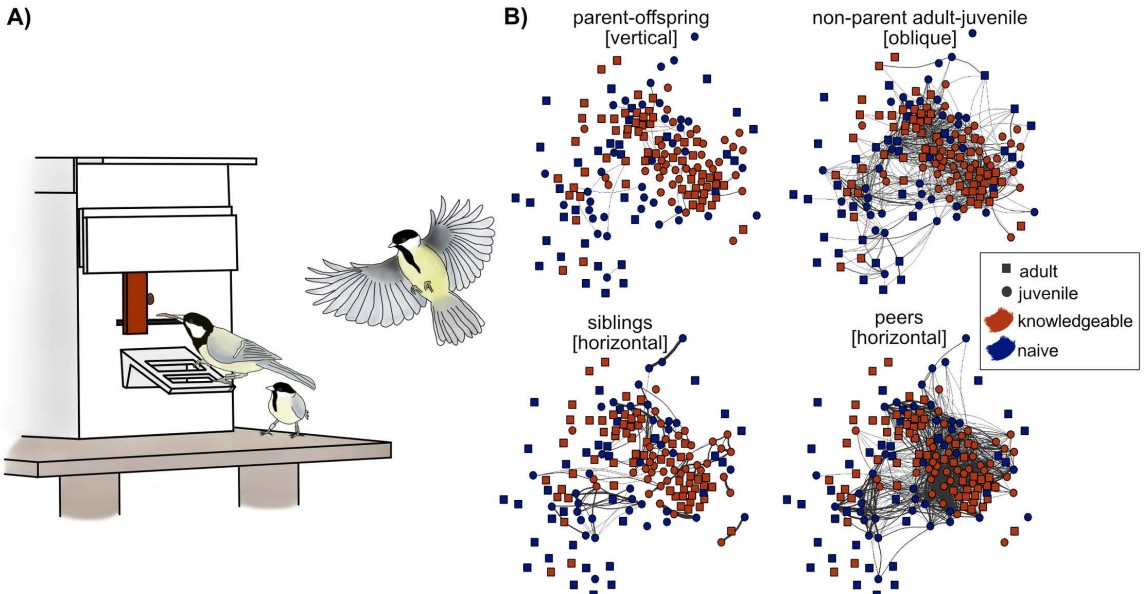

**Fig 1. Social learning pathways of learning to solve a puzzle box.** A) Fully automated puzzle box. Birds can access a mealworm tray by either sliding the door "left" or "right". B) Illustration of social network at MI depicting different potential pathways of learning. Adults are shown as squares, juveniles as circles with knowledgeable individuals in red and naïve individuals in blue. The data underlying Fig 1B can be found at https://doi.org/10.5281/zenodo.16930533.

the number of scrounges allowed us to distinguish between observational learning—i.e., offspring learn to solve because they are exposed to the motor-patterns—and a local enhancement effect, if simply being attracted to the puzzle box site by scrounging parents is sufficient for juveniles to learn to solve.

We found that juveniles whose parents produced no solves or scrounges during dependence had a probability of 0.11 [95% CI: 0.04–0.24] of becoming knowledgeable at solving. The odds of learning to solve increased by a factor of 274.84 [12.66–15,116.01] per magnitude increase in parental solves during dependence but were not significantly influenced by the number of parental scrounges, although a potential negative trend was apparent (odds ratio (OR): 0.49 [95% CI: 0.14–1.03] (S1 Table).

## Establishing social learning pathways of solving

To estimate the strength of social transmission and investigate pathways of the diffusion of solving behavior, we ran a multinetwork NBDA (S2 Table) [39] with four different social networks reflecting different pathways (Fig 1B): (1) a parent-offspring network to allow for vertical learning; (2) an oblique network allowing for social transmission from non-parental adults to offspring; (3) a horizontal network allowing transmission among siblings; and (4) a horizontal network allowing transmission among nonsibling peers. All networks were weighed by the dyadic association strengths based on cofeeding events at network feeders (see methods). We then modeled how closely the diffusion data—here the time at which juveniles learned to solve the box—followed the different networks, or combinations thereof. We included two individual-level variables (ILVs) with potential effect on the juveniles' learning rate [39], namely the number of parental solves produced during dependence, as well as the number of scrounges the juvenile produced up to the time of acquisition, as scrounging might facilitate future learning [42].

We found most support for social models with vertical transmission from parents, oblique transmission from adults, and horizontal transmission among siblings (summed Akaike weights ($\sum w_i$) = 0.65; S1 Fig). In the best performing model ($\sum w_i$ = 0.22), the social transmission rate was highest among siblings with 42.1% [95% CI: 41.5%–43.9%] of acquisition events, followed by oblique learning with 37.8% [95% CI: 32.2%–38.3%], and vertical learning with 19.8% [lower 95% CI: 14.0%] occurring as a result of the connections within the respective network (Fig 2A). Note that upper confidence intervals for the vertical social transmission parameter could not be obtained due to convergence issues in the best-performing model (likely due to a small number of acquisition events at some sites). However, the lower bound of the confidence intervals provides a reasonable lower limit for the social transmission effects in this case. The asocial learning rate in the best model was estimated at 0.3%, suggesting that no juveniles learned to solve the puzzle box through individual exploration.

While there was variation in who juveniles learned the solving behavior from, there was a distinct pattern in role models when comparing the first learners of a sibling cohort versus subsequent sibling learners. Of the first learners of each sibling cohort, 74.6% [median] were estimated to have learned through oblique social learning from nonparent adults and 23.6% vertically from parents. Of the subsequent learners in each sibling cohort, 93.9% were estimated to have learned from their siblings, 3.6% from nonparent adults and 1.6% from parents (Fig 2B). Neither of the ILVs tested (number of parental solves, number of scrounges produced by juveniles) had any effect on the social or asocial learning rate (all $\sum w_i$ < 0.5; S4 Table).

## Side choices

In a final set of models, we investigated the decision-making strategies that influenced juveniles' side choices (left or right), and how these strategies changed with increasing experience. As a proxy for social information available to juveniles, for each solve, we extracted the proportion of solves occurring on the right side in the five minutes immediately prior to solving. While there was evidence for local biases of solves occurring either on the left or right, juveniles were exposed to a wide range of social information on side choices (S2 Fig). We first tested the predictors of side choices on the first

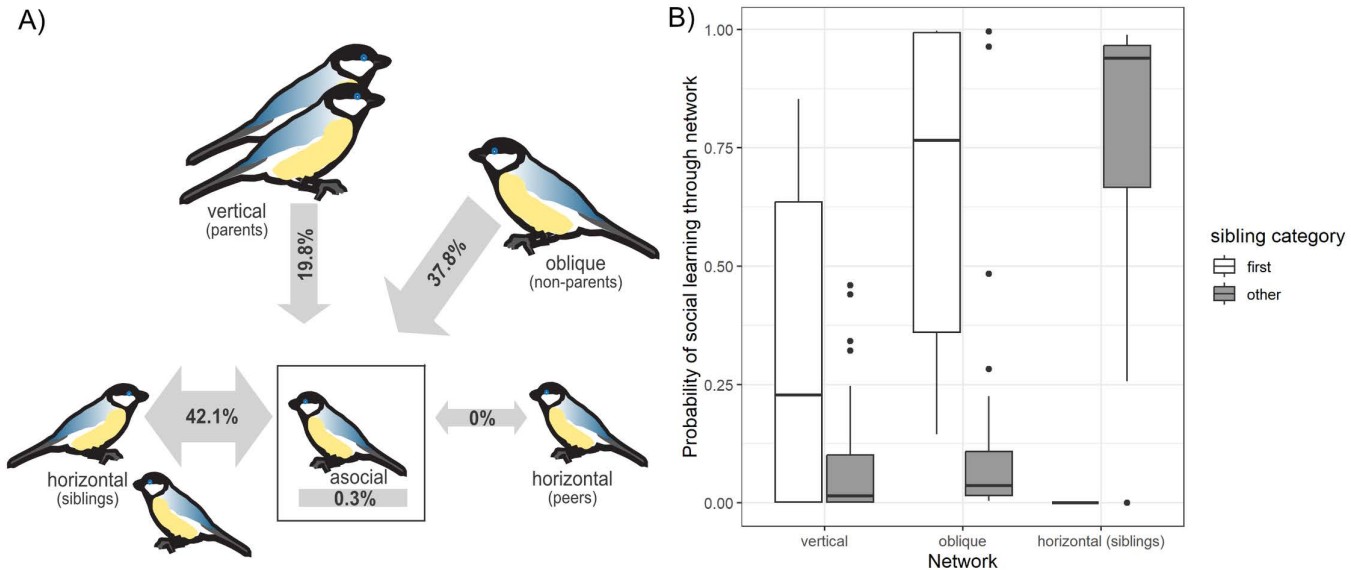

**Fig 2. Probability of learning through different pathways. A) Estimated percentage of acquisition events occurring as a result of vertical, oblique, horizontal, and individual social learning.** Social transmission was estimated to be strongest among siblings, followed by social transmission from nonparental adults (oblique) and parents (vertical), with no evidence for transmission among peers or asocial learning. **B) Estimated probability of learning through different pathways for first and subsequent learners within a sibling cohort.** First siblings learned primarily from parents and nonparent adults, while for subsequent siblings within a cohort, social learning was most likely to occur from the already knowledgeable first sibling. The data underlying this figure can be found at https://doi.org/10.5281/zenodo.16930533.

day of solving, when juveniles produced an average of 4 solves (range 1–51), totaling 211 solves. Our results showed that parental side choice during dependence did not predict juveniles' side choices on their first solving day (S5 Table; Fig 3A).

Instead, on the first solving day, the side choices of juveniles were strongly predicted by the side choices expressed by other birds immediately prior to solving, with the likelihood of producing a right solve increasing by a factor of 19.50 [95% CI: 5.84–68.62] when all solves by other birds occurred on the right compared to all occurring on the left. This was independent of the number of solves juveniles observed in the five minutes prior to solving (OR: 2.39 [95%: 0.72–8.62] per solve; S6 Table; Fig 3B).

Finally, to investigate how reliance on social and personal information might change with increasing experience, we considered 8,041 solves by 53 juveniles who had both social and personal information available at the time of solving. Overall, side choice was strongly predicted by personal experience (OR: 7.30 [95% CI: 5.07–10.49]), and reliance on personal information increased with increasing experience (OR: 1.22 [95% CI: 1.04–1.44] per 100 solves increase in experience; S7 Table; Fig 3C). We found a sizeable effect of social information influencing overall side choice (OR: 4.15 [95% CI: 2.89–5.98]), but juveniles decreased their reliance on social information with increasing experience (OR: 0.76 [95% CI: 0.67–0.86] per 100 solves increase in experience; S7 Table; Fig 3D).

## Discussion

Using fully automated foraging puzzles that allowed us to manipulate the knowledge state of breeding adults, we conducted a comprehensive investigation into the ontogeny of social learning in juvenile great tits, disentangling emergent patterns from detailed mechanisms and learning pathways. We found that the knowledge state of juveniles was strongly predicted by that of their parents, with offspring of parents that produced more solves during dependence being significantly more likely to learn to solve. Meanwhile, parental scrounging behavior during dependence did not predict the

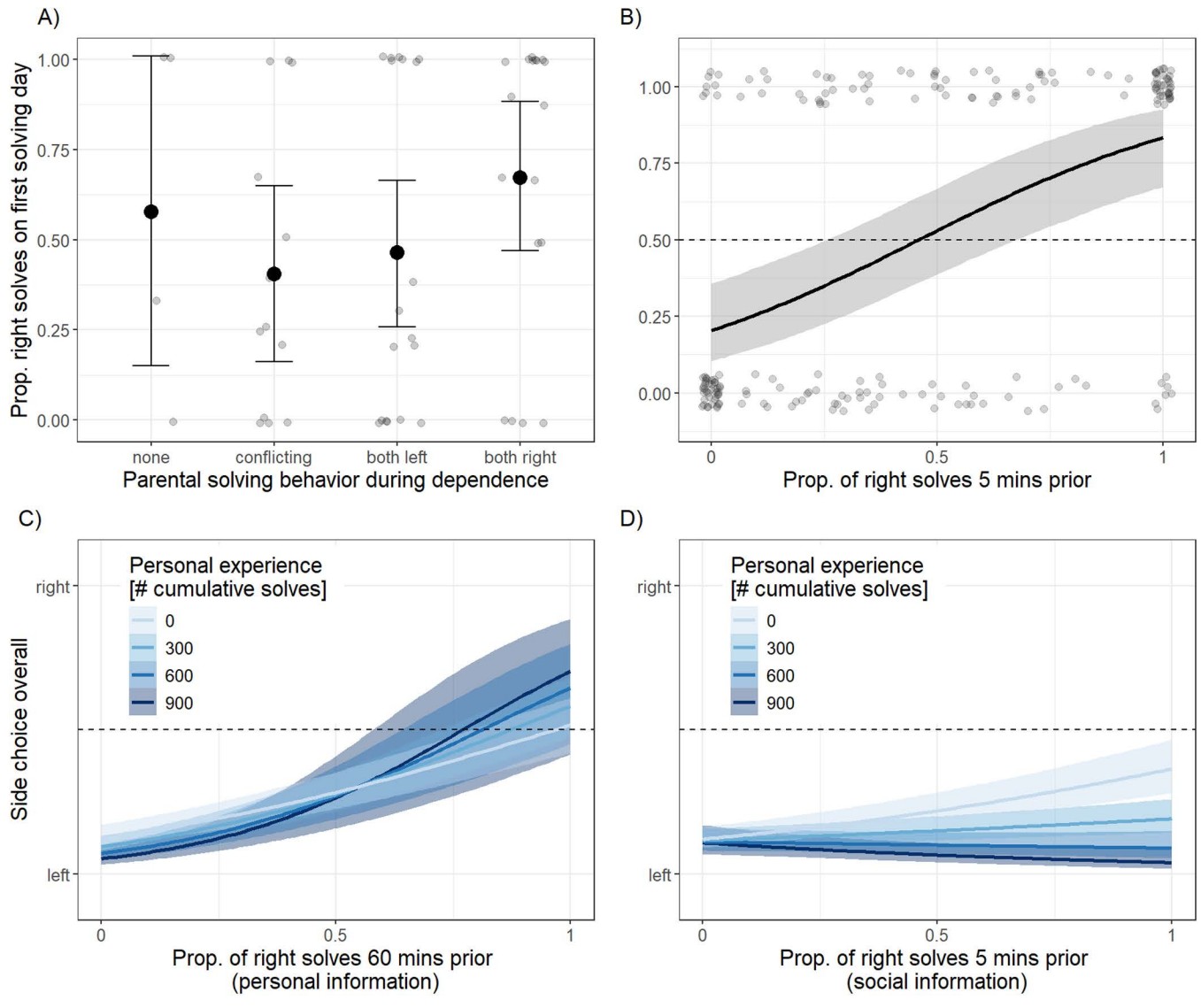

**Fig 3. Predictors of solving behavior of juveniles on their first solving day (A, B) and across all solving days (C, D), including 95% credible intervals.** Note that A,B show raw data (side choices on first day) as points. C,D are restricted to solves where juveniles have both social and personal data available (considering solves across all days). A) Influence of parental solving behavior during dependence on first-day side choices: The proportion of right solves produced by juveniles on their first solving day was not predicted by the type of information they received from parents during dependence, regardless of whether it was conflicting or nonconflicting. B) Influence of social environment on first-day side choices: Side choices by juveniles were predicted by side choices observed within 5 min prior to each solve. Juveniles were more likely to pick the side that was preferred by the majority of prior solving birds. C) Influence of personal experience on decision-making across all solves: Juveniles increased their reliance on personal information with increasing experience (measured in cumulative number of solves produced). D) Social influence on decision-making across all solves: While there was moderate evidence for an overall reliance on social information for side choices, juveniles reduced their reliance on social information with increasing experience. The data underlying this figure can be found at https://doi.org/10.5281/zenodo.16930533.

offspring probability of learning to solve, indicating that local enhancement to the puzzle site alone is not sufficient to acquire knowledge but instead suggestive of an effect of observational learning.

Yet, our detailed network-based diffusion analyses revealed that the acquisition of knowledge did not primarily occur through vertical transmission from parents, but rather from a mix of role models, including parents, nonparent adults, and

siblings. Strikingly, there was a difference in role models when comparing first and subsequent learners of each sibling cohort. While the first learners in each sibling cohort learned primarily from nonparental adults, subsequent learners within the sibling cohort almost exclusively learned from their already knowledgeable siblings. Meanwhile, only few seemed to have learned directly from parents. Our results provide a dramatic contrast from most previous studied systems exhibiting multigenerational cultures, where cultural transmission has been shown or assumed to overwhelmingly occur from parent to offspring, and give insights into the more variable transmission routes might occur across social learning species.

First, our results provide insights into potential alternative underlying mechanisms to vertical transmission that can lead to diet similarities observed between parents and offspring in passerines [34,35]. Our study suggests that early exposure to the foraging behavior of parents during dependence may facilitate learning, likely through observational learning of general motor patterns. Yet, full acquisition of foraging techniques may largely occur after juveniles have reached independence from their parents but are still associating with siblings [29]. This extends findings on diet similarities between cross-fostered passerine chicks and their foster parents [34,35], suggesting that these similarities are likely only in part a result of direct copying of parents, and social transmission among siblings is perhaps a more important driver of behavioral similarities within family units. Limited cultural inheritance of behavior from parents is further supported by the fact that for solving, the parental side choices during dependence did not influence juveniles' learning of side choices, regardless of parental experimental condition. In addition, the number of parental solves produced during dependence did not increase the speed with which juveniles learned to solve. The latter finding is consistent with a similar study on juvenile hihi, where juveniles exposed to experimental feeders during dependence by parents were not quicker to start to use them [18].

Second, our results highlight the importance of siblings as potential sources for social information in species with multiple offspring. This has been repeatedly demonstrated in humans, e.g., [43,44]. Among nonhuman animals, however, only few studies have investigated the role of siblings as sources for social information during ontogeny [45–47]. In line with our results, a study on captive juvenile ravens found that during experimental demonstrator–observer trials, siblings spent significantly more time in close proximity compared to nonsiblings and were much more likely to copy siblings' behavior [47]. Yet, to our knowledge, no studies have explicitly demonstrated the importance of siblings as role models in the wild in their natural social environment. Previous work at our site has found that great tit juveniles tend to stay in close association with their siblings for some time even after reaching independence from parents [29]. Our current study extends these findings, suggesting that the close relationships among siblings early in life are crucial for obtaining social information. Interestingly, in contrast with these results, there was little evidence for social learning of behaviors from nonsibling peers. The reason for this needs more investigation, but may be related to a preference or variation in tolerance between siblings and nonsiblings, or to preferences towards copying individuals most similar in developmental stage.

Finally, our findings provide insights into the dynamics of decision-making strategies across development. On their first solving day, side choices of juveniles were strongly predicted by the side choices of other birds immediately prior to solving themselves, disproportionally copying the majority [22]. Such conformist tendencies have previously been reported in great tits [27,48], as well as other passerines [18]. Yet this initial reliance on social information decreased rapidly with increasing experience, with juveniles showing a strong overall preference for personal over social information. These findings align with results from a study on captive great tits that found that after initial conformist social learning, individuals preferred personal over social information until experiencing changes in their social or physical environment [28,36]. This observed pattern is consistent with a strategy to learn socially when uncertain [22], and is a commonly found strategy in nonhuman animals (e.g., Norway rats [49], chimpanzees [50] or bumblebees [51]). Our findings on a strong reliance on personal information in juvenile great tits stand in contrast to a study on juvenile hihis, where initial side choices of entering a feeder were random, but subsequent visits appeared to be socially mediated with juveniles updating their preferences to match their peers' side choices through conformity [18].

Overall, our study provides experimental insights into the ontogeny of social learning and cultural inheritance in a species with limited parental care. We found that parental knowledge state strongly predicted knowledge acquisition of

offspring, but that exact foraging techniques were shaped by social information available at the point of learning and then crystallized over time with personal experience. More specifically, we demonstrate that offspring rely on a mix of role models, with our models highlighting the importance of siblings as important sources for social information. This suggests that in species that exhibit a short dependency period, cultural inheritance of information is first shaped by parents and then refined by later social interactions. This has implications for the persistence and patterning of cultures in such species, because if cultural knowledge is spread via multiple pathways, including oblique and horizontal learning, then cultural behaviors may be less vulnerable to extinction [52] compared to species with extended and exclusive parental care, where cultures might be restricted to specific maternal lines [53,54].

Future studies on learning during ontogeny should continue to investigate the various extrinsic and intrinsic factors that have been found to affect social learning and decision-making strategies in other species, including environmental factors (e.g., blue jays [55]), species-specific dispersal patterns (e.g., vervet monkeys [56]), personality traits (e.g., guppies: [57], barnacle geese: [58]), or physiological stress experienced during dependence (e.g., zebra finches: [59]).

## Materials and Methods

### Ethics statement

The use of animals adheres to the guidelines set forth by the Association for the Study of Animal Behaviour [60]. Birds were ringed under licenses held by LMA, GAN, and SW granted by the Radolfzell Bird Observatory (55-8841.03; 8853.17), and ethical approval was granted to LMA by the Regierungspräsidium Freiburg (35-9185.81/G-19/159). Participation in experiments occurred on a voluntary basis.

### Field methods

**Study site and population.**  The study took place at the Max Planck Institute of Animal Behavior in Radolfzell, Germany, at two mixed woodland field sites (MI: 47.76555, 8.99757; GU: 47.77463, 8.99937 [WGS84]). Great tits were caught in mist nets and equipped with a uniquely numbered metal leg ring issued by the Radolfzell Bird Observatory, and a PIT tag (Eccel Technology, EM4102) for individual identification. To study their breeding behavior, we provided 207 nest boxes (Schwegler type 1B, 2M, 3SV) at MI, and an additional 99 (Schwegler type 2M) at GU. Throughout the breeding season in 2021 from early April until late June, we monitored all nest boxes on a regular basis to record any breeding activity. Four days after hatching of nestlings, we identified the breeding pair by replacing the front plates of the nest boxes with faceplate loggers (NatureCounters Ltd.) that recorded PIT tags of adult birds entering through a built-in RFID antenna. We subsequently caught untagged breeding adults in the nest box and equipped them with a leg ring and PIT tag. At 15 days of age, we equipped nestlings with a metal leg ring and a PIT tag. To determine the exact date of fledging, we completed daily nest checks from 22 days after hatching until all nestlings had fledged.

**Puzzle box experiments and network data collection.**  To conduct our social learning experiments, we used fully automated puzzle boxes, with the design based on a sliding-door puzzle that has previously been used in social learning studies on great tits [27,36,37] (Fig 1A). Birds could slide a door either to the left or right to obtain a mealworm reward, with an RFID antenna built into the perch of the box registering the landing bird's PIT tag. The puzzle boxes were made selective, i.e., in that access to "left," "right," or "both" sides could be granted based on each bird's individual PIT tag [37].

Approximately 2 weeks prior to fledging of the first nestlings, we deployed four puzzle boxes at MI and three at GU. Sub-populations at both sites contained previously knowledgeable birds who had learned to solve the puzzle in prior, independent experiments [37]. After letting the behavior diffuse from these knowledgeable birds for 2 weeks, we restricted access for all knowledgeable adult birds with a minimum of 10 solves to either "right" or "left". This resulted in four different experimental conditions, namely (1) breeding pairs with both breeding partners restricted to the same side (either both right or both left), (2) breeding pairs with partners restricted to opposite sides (one right, one left), (3) breeding pairs where only one partner was knowledgeable (either right or left), and (4) breeding pairs without any knowledge of how to solve

the puzzle box. We then let the behavior diffuse throughout the breeding season until the 9th of July 2021, recording identity and side choices of all solving birds—both adults and juveniles as they entered the population. In each experimental week, we deployed puzzle boxes on 4 days, after which they were removed for data download, repair, and cleaning.

On two of those remaining three days each week on which puzzle boxes were removed, we installed openly accessible feeder columns at the seven puzzle box locations containing a mix of sunflower seed and kibbled peanuts. The feeding holes of these feeders were fitted with two receiving RFID antennae (NatureCounters Ltd.), which recorded visiting birds' PIT tags (logger: Priority 1 Design). These visits were used to track associations among individuals and calculate social networks (see below).

## Statistical methods

All analyses were run in R4.4.0 [61]

**Data preparation.** For all statistical analyses, we considered juveniles and adults as knowledgeable if they had solved the puzzle box at least 10 times. We used a more conservative threshold compared to previous studies that considered birds as knowledgeable after three solves [27,62], as we found that the RFID antenna on the box sometimes failed to correctly identify the solving bird if several birds perched on the antenna simultaneously, which occasionally occurred among juveniles in particular due to high social tolerance.

**Influence of parental behavior during dependence on probability of solving.** In a first analysis, we investigated to what extent offspring behavior is predicted by vertical transmission from parents during dependence, specifically testing whether more frequent exposure to solving behavior during dependence increases the probability of juveniles learning to solve. We ran a Bayesian logistic regression with a binary outcome variable—learned to solve or not—with the log number of solves and the log number of scrounges produced by parents between fledging and their offspring's date of acquisition of solving behavior as predictors. Inclusion of the number of scrounges controls for a local enhancement effect, if simply being attracted to the puzzle box site by scrounging parents is sufficient for juveniles to learn to solve. For juveniles who did not learn to solve, we used the number of parental solves produced up to the 15th day after fledging, which corresponds to the mean number of days until juveniles started to solve. To aid model convergence, we used weakly regularizing priors with normal distribution ($\mu = 0$, $\sigma = 10$) for intercept and ($\mu = 0$, $\sigma = 5$) for slopes, respectively. We ran four chains with 6,000 iterations each (2,000 for warm-up, 4,000 for sampling) in the brms package [41]. We conducted visual inspection of trace plots to confirm chain mixture, convergence, and stationarity, and a posterior predictive check [63]. Effect sizes were based on posterior means and confidence intervals on 95% credibility intervals [63]. We report OR for each variable which refer to the change in the odds of an event occurring per one-unit change in that predictor while all other predictors are held constant [64]. We interpret effects as significant if the confidence intervals of OR fall either above 1 for a positive effect or below 1 for a negative effect.

**Establishing learning pathways of solving.** To estimate the strength of social transmission and investigate potential pathways of the diffusion of the solving behavior, we ran NBDA with multiple diffusions for four out of the seven puzzle box locations [38–40]. Three sites were removed due to insufficient number of learners (S2 Table). NBDA infers social transmission if the spread of a behavior follows the social network [39]. We used the time of acquisition diffusion analysis variant of NBDA, which considers the time at which individuals acquire a behavior as diffusion data [38]. We considered the time of each knowledgeable bird's third solution (regardless of the side choice), as diffusion data to reduce noise in the case of any misidentification of solver identity.

To establish the potential pathways of transmission, we used four different networks in a multinetwork NBDA [3,65]: First, a parent-offspring network allowing for vertical social transmission from parents to offspring. Second, an oblique network allowing social transmission from nonparent adult individuals to offspring. Third, a horizontal social network measuring social transmission among siblings. Fourth, a horizontal social network allowing for social transmission among nonsiblings. All networks were weighted by dyadic association strengths based on cofeeding events at network feeders (see above).

To exclude transient individuals and increase certainty about the strength of birds' social connections with others within the social network, we chose to only include birds in the model that had been registered at the RFID feeders at least 10 times. From the spatiotemporal data at these feeders, we used Gaussian mixture models to identify clusters to assign individuals to visiting groups [66]. We then used the gambit of the group approach [67] to calculate association strengths using the simple-ratio index, which ranges from 0 (never seen together) to 1 (always observed in the same group) [68,69].

As associations with other birds change over time [29], we used a dynamic framework in which we recalculated the social network at each acquisition event based on the association data collected in the week before and after each event. To further account for the fact that birds can enter (e.g., new birds hatching) or leave (e.g., through death or emigration) the study population, we included a presence matrix [39]. For each acquisition event, we extracted whether a bird had been present during the previous seven days, either registered on the network feeders or on the puzzle box at the respective location. If present at the respective puzzle box site, birds were assigned an entry of 1, if absent, they were assigned an entry of 0. As such, birds could only learn and transmit the behavior if they were present at the respective location.

The diffusion was set to start on the day the first juveniles entered the population (14th of May 2021). All adult birds that had learned to solve the box prior to that date were included in the models as demonstrators [39]. Any adult bird that learned to solve the puzzle box during the diffusion, i.e., after the 14th of May, was filtered out, meaning that the diffusion was modeled without them, but juveniles could still learn from them after they acquired the behavior [39]. To account for the fact that high-frequency solvers may be more likely to transmit behavior, we included transmission weights by calculating the rate at which they were performing the behavior per experimental day after they had acquired it.

We additionally included two individual-level variables (ILVs) with potential influence on a juvenile's social and asocial learning rate [39]. First, we extracted the number of solves produced by parents since fledging until juveniles acquired the behavior as a proxy for exposure to the puzzle box solution during dependence. For juveniles that did not learn to solve, we extracted the number of parental solves from fledging until the juveniles were 15 days old, which corresponded to the average time period until knowledgeable juveniles started solving. Second, we extracted the number of scrounges each individual had performed up until each acquisition event (defined as visits within 2 s after a solve when the door was open), as scrounging represents an alternative strategy that might facilitate future learning [42]. All ILVs were included as time-varying, meaning the numbers were updated at every acquisition event. For better convergence, ILVs were centered around 0. We fitted unconstrained models, with ILVs being allowed to influence asocial and social learning rate independently. The baseline rate of asocial learning was set to an average individual with ILVs set to 0 [39].

We used NBDA version 0.9.6 [70], and following procedures suggested in [39], we fit the model set with two different baseline rates ("constant" and "Weibull") in all combinations of the four networks and the two ILVs, both with and without social transmission. This resulted in 256 models for each baseline rate. Since models with a Weibull baseline rate showed more support ($\sum w_i = 0.86$) compared to those with a constant baseline rate , we only considered models with the better performing baseline rate (Weibull) for inference. We calculated network and variable support using the Akaike Information Criterion corrected for sample size (AICc) [71]. We additionally extracted effect sizes as model averaged medians for all supported variables with summed Akaike weight ($\sum w_i$) >0.5, and 95% confidence intervals based on the best performing model in which they occurred using profile likelihood methods [39].

**Side choices.** In a last set of three models, we investigated the decision-making strategies that influenced juveniles' side choices (left or right), and how these strategies changed with increasing experience.

The first two models focused on the juveniles' solving behavior on their first solving day when they had no prior solving experience. First, in a Bayesian linear regression, we tested whether the juveniles' proportion of right solves on their first solving day was predicted by the parents strategies during dependence as a categorical variable with four categories: "none" for nonsolving parents, "nonconflicting right" if more than 90% or parental solves occurred on the right side, "nonconflicting left" if more than 90% of parental solves occurred on the left side and "conflicting" if the proportion of parental solves on either side was less than 90%.

Second, we investigated whether side choices on the juveniles' first solving day were influenced by their social environment. In a Bayesian logistic mixed effects model, we tested whether a juveniles' side choice (right/left) was predicted by the proportion of right solves observed during the 5 min immediately prior to solving, while additionally allowing for an interaction with the total number of solves that were produced by others in the 5 min prior. We included the ID of the juvenile as a random effect.

In the third and final model, we considered solves across all solving days to investigate how the reliance on social and personal information changed with increasing experience. We therefore restricted the data to all solving events where juveniles had access to social information (minimum of one solve produced by another bird in the 5 min prior) as well as asocial information (juvenile solved at least once in the last hour). In a Bayesian logistic mixed effects model, we investigated whether side choice (right/left) was predicted by the proportion of right solves observed within 5 min prior to producing a solve (social influence) and the proportion of right solves a juvenile had produced within the last hour (personal experience). The 5-min window was chosen to be consistent with previous studies [27]. We additionally allowed for an interaction of both terms with the cumulative number of solves by each juvenile to test for changes in reliance on personal versus social information with increasing experience. We included the ID of the juvenile as a random effect.

In all three models, we used MCMC sampling with 4 chains of 4,000 iterations each (plus 2,000 for warmup) using the package "brms" [41]. We conducted visual inspection of trace plots and extracted R-hat values to confirm chain mixture, convergence, and stationarity, and conducted a posterior predictive check [63]. Finally, we extracted effect sizes and confidence intervals using posterior means and 95% credible intervals [63]. We again report ORs for each predictor variable.

## Supporting information

**S1 Fig. Summed Akaike weights across models with different transmission pathways for the diffusion of solving behavior.** NBDA showed most support ($\sum w_i = 0.64$) for models with social transmission between parents and offspring (network 1: vertical), between nonparent adults and juveniles (network 2: oblique), and among siblings (network 3: horizontal (siblings). Numerical values area available in S3 Table and at https://doi.org/10.5281/zenodo.16930533.
(TIFF)

**S2 Fig. Histogram of the proportion of right solves occurring in the 5 min immediately prior to juveniles producing a solve A) on the juveniles' first solving day and B) across all experimental days.** These values serve as a proxy for social information available to the juveniles at the time of solving. Data underlying this figure can be found at https://doi.org/10.5281/zenodo.16930533.
(PNG)

**S1 Table. Model output of parental behavior predicting behavioral strategy of offspring (odds ratios).**
(XLSX)

**S2 Table. Summary of data included in NBDA analysis.**
(XLSX)

**S3 Table. Summed Akaike weights across models for different transmission pathways.**
(XLSX)

**S4 Table. Support for ILVs in summed Akaike weights ($\sum w_i$) and effect sizes for supported individual-level variables (with $\sum w_i > 0.5$). .**
(XLSX)

**S5 Table. Pairwise comparison of estimated marginal means of parental strategy [none, conflicting, both left, or both right] during dependence predicting offspring side choices on first solving day.**
(XLSX)

**S6 Table. Influence of social observations on first day side choices of juveniles (odds ratios).**
(XLSX)

**S7 Table. Change of influence of personal experience and social environment with increasing experience (odds ratios).**
(XLSX)

## Author contributions

**Conceptualization:** Sonja Wild, Lucy M. Aplin.

**Formal analysis:** Sonja Wild.

**Funding acquisition:** Lucy M. Aplin.

**Investigation:** Sonja Wild, Gustavo Alarcón-Nieto.

**Methodology:** Sonja Wild.

**Project administration:** Sonja Wild.

**Visualization:** Sonja Wild.

**Writing – original draft:** Sonja Wild.

**Writing – review & editing:** Gustavo Alarcón-Nieto, Lucy M. Aplin.

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
