## [Editor Report · Decision Letter 0]

11 Jul 2025

Dear Dr Wild, 

Thank you for submitting your manuscript entitled "Experimental evidence reveals alternative pathways to cultural inheritance in juvenile great tits, Parus major" for consideration by PLOS Biology.

Your manuscript has now been evaluated by the PLOS Biology editorial staff, as well as by an academic editor with relevant expertise, and I am writing to let you know that we would like to send your submission out for external peer review.

We would like to consider your article as a Short Report. Please select this article type when you complete the next step of submission.

Once your full submission is complete, your paper will undergo a series of checks in preparation for peer review. After your manuscript has passed the checks it will be sent out for review. To provide the metadata for your submission, please Login to Editorial Manager (https://www.editorialmanager.com/pbiology) within two working days, i.e. by Jul 13 2025 11:59PM.

Kind regards,

Taylor

Taylor Hart, PhD, 

Associate Editor

PLOS Biology

thart@plos.org

---

## [Decision Letter · Decision Letter 1]

21 Aug 2025

Dear Dr Wild,

Thank you for your patience while your manuscript "Experimental evidence reveals alternative pathways to cultural inheritance in juvenile great tits, Parus major" was peer-reviewed at PLOS Biology. I'm handling your manuscript temporarily while my colleague Dr Taylor Hart is out of the office. It has now been evaluated by the PLOS Biology editors, an Academic Editor with relevant expertise, and by four independent reviewers. 

You'll see that reviewer #1 is very positive, and simply wants some minor textual changes and wonders if there’s an error in one Figure. Reviewer #2 is similarly extremely positive and only has one minor suggestion. Reviewer #3 is also very positive, and just has a list of minor textual changes. Reviewer #4 is again very complimentary, and just has a few textual requests.

Based on the reviews, we are likely to accept this manuscript for publication, provided you satisfactorily address the remaining points raised by the reviewers and the following data and other policy-related requests.

IMPORTANT - please attend to the following:

a) Please make your Title more explicit; we suggest "Juvenile great tits learn how to solve a foraging puzzle from siblings and other non-parental adults" or "Social learning in juvenile great tits occurs through contributions from siblings and other non-parental adults"

b) Please address the points raised by the reviewers.

c) Please address my Data Policy requests below; specifically, we need you to supply the numerical values underlying Figs Figs 1B, 2AB, 3ABCD, S1, either as a supplementary data file or as a permanent DOI’d deposition. I note that you already have an associated GitHub deposition (https://github.com/sonjawild/fledgie_learning). Please could you confirm whether this is sufficient to recreate the Figures? Also, because Github depositions can be readily changed or deleted, please make a permanent DOI’d copy (e.g. in Zenodo) and provide this URL (see below).

d) Please cite the location of the data clearly in all relevant main and supplementary Figure legends, e.g. “The data underlying this Figure can be found in S1 Data” or “The data underlying this Figure can be found in https://zenodo.org/records/XXXXXXXX

e) Please include the URLs of your funders in the Financial Disclosure statement.

We expect to receive your revised manuscript within two weeks. 

*Published Peer Review History*

*Press*

Sincerely,

Roli Roberts 

Roland G Roberts PhD

Senior Editor

PLOS Biology

rroberts@plos.org

on behalf of

Taylor Hart, PhD, 

Associate Editor

thart@plos.org

PLOS Biology

DATA POLICY:

Regardless of the method selected, please ensure that you provide the individual numerical values that underlie the summary data displayed in the following figure panels as they are essential for readers to assess your analysis and to reproduce it: Figs Figs 1B, 2AB, 3ABCD, S1. NOTE: the numerical data provided should include all replicates AND the way in which the plotted mean and errors were derived (it should not present only the mean/average values).

CODE POLICY

DATA NOT SHOWN?

REVIEWERS' COMMENTS:

Reviewer #1:

[identifies themselves as Takao Sasaki]

These authors investigated social learning strategies in juvenile great tits and found that oblique transmission—learning from non-parental individuals—is stronger than vertical transmission from parents. Interestingly, their data also show that juveniles were highly influenced by informed siblings.

The manuscript is exceptionally well-written, and the findings are of broad interest to researchers across multiple disciplines.

I have only a few minor comments.

1) The first part of the Results section includes detailed methodological descriptions, but I wonder if this could be made more succinct. While I understand the Methods section is at the end of the manuscript and the authors wanted to provide context, I suggest focusing on only the essential information needed for readers to interpret the results.

2) Figure 3B: Is the x-axis should be "Prop of right solves 5 mins prior" instead? Also, it seems the data points do not seem to match with the fitted curve. i.e. there are many "right" choices when x = 0 and many "left" choices when x = 1, suggesting that the fitted curve should actually be reversed. I wonder if the authors accidentally switched "right" and "left" on the y-axis (as the fitted curve seems correct). 

Reviewer #2:

This paper makes a novel and important contribution to the currently highly active field of animals' social learning and culture, frequently featured in this journal. Research on intergenerational cultural transmission has tended to focus on 'k-selected' species with long periods of juvenile dependence on parents, such as most primates and cetaceans. These also often involve a single offspring.

By contrast many other, 'r-selected' species have only a short period of dependence on parents, and there are multiple simultaneous siblings. Great tits are a good example. 

This rigorous experimental study of wild great tits' learning during ontogeny clearly shows that social learning is crucial for the foraging skill required (none learned to solve the task if limited to individual exploration), but that while the first learners among siblings tended to learn from parents, or even more so from other adults, later learners tended to learn preferentially from their siblings. Figure 2 offers a nice summary. The more detailed aspect of matching the side of the foraging device observed was more dependent on the most recently observed models, rather than that preferred by the parents. All these findings are new to the field and importantly expand our understanding of the alternative pathways of cultural transmission found across the animal kingdom.

The manuscript was a delight to read in its clarity and the figures are also very clear and helpful. I judge the methods and statistical analyses to be rigorous and appropriate. The coverage of the literature in the Introduction and Discussion sections is appropriate and both are well judged.

Accordingly I have only the most minor of suggestions for a slight revision: could it be briefly explained why there were only two cases with one parent knowledgeable and one parent ignorant? (line138). So no analysis of the effects of this could sensibly be made.

In many years of reviewing I don't recall making a more positive recommendation.

END 

Reviewer #3:

Thank you for the opportunity to review this manuscript.

I judge this a well-conducted study that should be of substantial interest to the field of animal culture, and I also judge it to be appropriate for this journal. Here, the authors build on previous work concerning the spread and maintenance of puzzle box-solving behavioural traditions in great tits (Parus major). 

The box-opening frequency of the parents during the dependent period did predict whether juveniles would learn to open the boxes, but other birds within the social environment (especially siblings) turned out to be of greater importance, especially once one sibling in a family group had learned from the parents. The actual method for box opening (pushing left or right) was also influenced by other birds within the social environment: siblings, and non-parental adults. Providing this level of detail about the spread of the behaviour is really fascinating, and the implications of this are discussed by the authors.

Overall, the study is well-presented, and I have relatively minor suggestions for revisions: these mostly pertain to clarity of expression/wording. 

56: "…can greatly increase the efficiency with which individuals acquire novel information or behaviour, often leading to a concentration of social learning during the juvenile period [10]" - 

This logic is a little unclear. Would it be better to say: …can greatly increase the efficiency with which individuals acquire novel information or behaviour. Close proxmity to parents and siblings often leads to a concentration of social learning during the juvenile period [10]?

64: "This then in leads…" - typo? Would it be better to read "This then in turn leads to…"

66: "Yet, social learning is not restricted

to species with prolonged dependence periods, especially in many bird species [17]" - the wording is a little clunky. How about "However, social learning is not restricted to species with prolonged dependence periods: this is particularly true of birds [17]" - and then perhaps a short addition clarifying that many bird species have short periods of dependence relative to their lifespan, or something, to help build the picture?

76: Maybe expand a little? e.g. "… [20,21]. It may be better to learn from individuals they encounter in these new environments." (or something to this effect)

88: Would it be more accurate to say "as multi-generational cultural traditions"?

94: "We currently lack information on…" might be better wording.

100: "With onset of…" - would it be better to say "With the onsent of…" or "On the onset of…"

101: "Previous work used cross-fostering experiments…" - possibly reword to "Previous cross-fostering experiments, where eggs were swapped between great tits and blue tits (Cyanistes caeruleus), found that…"

105 onwards: Do you mean 'vertical learning' solely to refer to parent -> offspring social learning, and no other form of learning? It might be worth clarifying "…and subsequently copied behaviour from other individuals in their foster species after reaching independence, but not necessarily from their foster parents."

135 "Solver's identifies were identified" - maybe "Solver's identities were determined" would be better to avoid redundancy.

140 Would "the behaviour continued to diffuse" or "we allowed the behaviour to continue to diffuse" be more accurate?

163 Local enhancement, to my understanding, is still a form of social learning (and so would not preclude the behaviour from being culturally transmitted?). What would the implication of a local enhancement effect be for these results? Perhaps this just isn't clear to me why local enhancement should be controlled for in this section - unless you are specifically wanting to determine that the learning must be observational. I would suggest making this clearer/justifying it. 

164 onwards:

I know this is said in the methods (birds were considered knowledgeable after three solves) but this could use defining here as well as in the methods 

195: most of the first siblings to learn, learned from non parent adults! subsequent learners mostly learned from siblings.

211: Would this be better as "In a final set of models" or "In the last set of models"?

245: Should this be "manipulate the knowledge state"?

255: Should this be "difference in" or "distinction between"?

256 onwards: oblique learning from non-parental models seemed to be higher than from parental models (line 185 onwards) - but this isn't really discussed at all. I think this should be highlighted and considered in the discussion, too. 

267: Is this a feature common to great tits? In general, I think there could be some more description of great tit background behaviour/life history, possibly in the introduction. i.e., you mention this behaviour from line 284 onwards, but it might be better in the introduction (to build the story behind this work). 

Reviewer #4:

This study presents experimental evidence for the drivers of information use and foraging decisions in newly-independent songbirds. This study is initially similar in outlook to a previous work (i.e. studying information use and foraging decisions during early life in a short-lived passerine with limited parental care - see Franks et al. 2020); however, the present study offers an elegant and more extensive study design to develop this area further, and I would like to commend the authors for this. I also found this paper very clear to read. As a result, I have only a few minor comments to raise.

L143: How were the different solutions mixed across the population to ensure juveniles would encounter a variety of solutions once fledged - i.e. how do juveniles disperse across the study site? Perhaps just worth a mention here or in Methods.

L265-266: I'm just wondering about the mechanism here - is this likely to be a learned aspect, e.g. experienced more feeder use with parents, or more innate e.g. inheritance of traits which mean they are more likely to interact with feeders (and thus potentially solve)? Could a small discussion of this be added?

L275 and L303: correct "hihis" to "hihi" - the plural works in the same way as "sheep"!

L302: I think the vervet monkey study [50] showed that migrant males continued to prefer the dominant food colour later on too - i.e. it persisted after their initial choice, even with some personal experience (in that paper's Supplementary Information). There, conformity was (a) "a disposition to copy the choices of a majority of others in a social group", and (b) "a willingness to subjugate one's own countervailing knowledge in matching the majority's choice". So perhaps this doesn't quite reflect what you found, if great tits were less likely to conform later on?

---

## [Editor Report · Decision Letter 2]

5 Sep 2025

Dear Dr Wild,

Thank you for the submission of your revised Short Reports "Siblings and non-parental adults provide alternative pathways to cultural inheritance in juvenile great tits" for publication in PLOS Biology. On behalf of my colleagues and the Academic Editor, Lars Chittka, I am pleased to say that we can in principle accept your manuscript for publication, provided you address any remaining formatting and reporting issues. These will be detailed in an email you should receive within 2-3 business days from our colleagues in the journal operations team; no action is required from you until then. Please note that we will not be able to formally accept your manuscript and schedule it for publication until you have completed any requested changes.

PRESS

Sincerely, 

Taylor Hart, PhD, 

Associate Editor

PLOS Biology

thart@plos.org